# Growth Priorities of the Four Riverine Cyprinids during Early Ontogeny

**DOI:** 10.3390/ani13142345

**Published:** 2023-07-18

**Authors:** Krzysztof Kupren, Joanna Nowosad, Dariusz Kucharczyk

**Affiliations:** 1Department of Tourism, Recreation and Ecology, Institute of Engineering and Environmental Protection, Faculty of Geoengineering, University of Warmia and Mazury in Olsztyn, 10-719 Olsztyn, Poland; 2Department of Ichthyology, Hydrobiology and Aquatic Ecology, National Inland Fisheries Research Institute, Oczapowskiego 10, 10-719 Olsztyn, Poland; nowosad.joanna@gmail.com; 3Department of Research and Development, Chemprof, Gutkowo 54B, 11-043 Olsztyn, Poland; darekk56@gmail.com

**Keywords:** allometric growth, chub, common barbel, common dace, early development, ide

## Abstract

**Simple Summary:**

The present study analyzed growth priorities during early development in four riverine cyprinid species. The allometric growth of the species studied is in line with the growth pattern typical of many fish species, where high allometric (rapid) growth of head and tail sections is observed at the beginning of the larval period and a clear tendency towards isometric (more balanced) growth at the end of the larval period. It was also observed that a lower level of ontogenetic development of fish after hatching was associated with higher growth rates during the first days after hatching. Specific to the species studied was the absence of abrupt changes in body proportions and the persistence of positive allometric growth in body size throughout the larval and juvenile periods, most likely related to a gradual change in behavior and habitat.

**Abstract:**

Growth patterns during early development of four species of riverine cyprinids, common dace (*Leuciscus leuciscus* L.), ide (*Leuciscus idus* L.), chub (*Squalius cephalus* L.) and common barbel (*Barbus barbus* L.), over 30 days of rearing, were analyzed. The study period covered three successive phases of larval development (preflexion, flexion, postflexion) and part of the juvenile development. The growth analysis carried out in the present study shows that significant allometric growth occurred mainly in preferentially localized body parts (head and tail) and was also associated with an increase in body height. The replacement of temporary structures (e.g., fin fold absorption) and the appearance of definitive structures around 21 mm TL (beginning of the juvenile period) corresponds to a stabilization of the relative growth of most parts of the body. This is probably due to the fact that the studied species had completed the most important part of the remodeling process (metamorphosis) and had achieved a body shape that allows juveniles to colonize faster water habitats. The results also indicate that individuals of species that are less developmentally advanced at the time of hatching (common barbel, chub) show greater initial dynamics of change in growth rate than individuals of species whose individuals are more advanced in ontogeny (common dace, ide). In addition, the small differences observed in growth parameters between successive periods analyzed, supported by the results of previous studies on these species, probably reflect the lack of abrupt changes in the behavior and habitat of the species studied.

## 1. Introduction

During early development, fish embryos and larvae undergo rapid and extensive morphogenesis and differentiation [1], including changes in body proportions and physiology and, consequently, in behavior [1,2]. These changes have been used to divide the larval period into periods, phases, and stages, each with different morphological, physiological and ethological capabilities that define larval performance throughout development [3,4,5]. These drastic changes in body shape that occur during the larval period are the result of differential growth rates of some body segments with respect to total length (TL), also known as allometry [6,7]. The presence of allometric growth is a diagnostic feature of fish larvae, and its presence has been considered an adaptation to environmental conditions that may affect their survival and growth [1,8]. During the early stages of fish development, the growth rate of body parts/structures involved in sensing, feeding, breathing and swimming is faster than that of the total body length. As the organs become relatively well developed and can effectively fulfil their role, their growth rate slows significantly or equals the growth rate of the whole body, ensuring that the most important organs of the fish can grow and develop preferentially [9]. Observed changes in the growth rate of individual body parts are usually determined by identifying inflexion points. The presence of inflexion points in the growth coefficients of body segments is a common feature in larval metamorphosis. It signals a shift in the direction of allometric growth, resulting in changes in the larval body shape [1,9,10,11].

The present study analyzed growth priorities during the early development of four riverine cyprinid species. Two of these, common dace (*Leuciscus leuciscus* L.) and ide (*Leuciscus idus* L.), are included in the genus *Leuciscus*. The chub (*Squalius cephalus* L.) is currently included in the genus *Squalius* (although its systematic status is not clearly defined, and many scientists still include it in the genus *Leuciscus*). The common barbel (*Barbus barbus* L.) is a member of the genus *Barbus*. They are rheophile species with sizes ranging from small, e.g., common dace (standard length up to 40 cm), medium, e.g., ide and chub (60–80 cm) to large, e.g., common barbel (standard length up to 120 cm). All these species are usually found in the middle and lower reaches of European rivers. During the juvenile and adult periods, these fish live in fast-flowing rivers, preferring places near the bottom (barbel) or in open water (ide, dace, chub) [12]. Adults are omnivorous with a very varied diet. These species spawn in early (common dace, ide) or late spring (chub, common barbel). Adults are non-guarding rock and gravel spawners with pelagic A.1.2 (common dace, ide) or benthic A.1.3 (chub and common barbel) larvae [13]. After hatching, larvae tend to migrate downstream from the spawning grounds to parts of the river with slower currents and high plankton productivity [12]. All species presented have been the subject of several detailed studies of early life stage biology and, as one of the few riverine cyprinids, an analysis of the dynamics of changes in body proportions. These show that they have very similar rearing requirements and despite their different sizes and slightly different behavior immediately after hatching, individuals of these species undergo gradual changes in body shape during larval development. A gradual change in body proportions in the species studied is reflected both in the absence of a shift in overall growth rates and in the absence of apparent coupling inflexion points in parameters that show bi-phasic growth (changes in growth curves that occur over a wide range of body lengths and ages). Such growth patterns are associated with the absence of sharp behavioral and habitat changes (observed under both laboratory and natural conditions) and may also be considered a gradual adaptation to life in flowing water [14,15,16,17].

The aim of this study is to present growth priorities during early morphological development of common dace, ide, chub and common barbel. Laboratory-reared specimens were studied from hatching to 30 days post hatching (DPH) to investigate allometric growth patterns during successive phases of larval and early juvenile life. The study tested the hypotheses that growth is prioritized in the most anterior and posterior parts of the body (i.e., head and tail), and that its dynamics and duration are influenced by the developmental stage of the hatching larvae and their behavior during successive days of development. This research will complement the above-mentioned studies on these specimens, describing early developmental stages by identifying inflexion points, and will allow for the first time a wider discussion of priorities in early fish development, taking into account in particular the existing single scientific reports in this area devoted to other cyprinid species.

## 2. Materials and Methods

The source of data for the analyses presented in this paper were the results obtained by the authors of this paper during research which formed the basis of previously published work on the early ontogeny of these species [14,15,16,17]. In all cases, the fish were spawned artificially using methods described earlier for common dace [18], ide [19], chub [20] and common barbel [21]. In all cases, the fertilization was carried out using the optimal method [22,23] and incubation was conducted under optimal environment conditions [24]. Furthermore, the rearing was carried out under optimal environmental and feeding (live food i.e., *Artemia* sp. Nauplii (San Francisco origin)) conditions. In this paper, the results of the measurements of selected morphometric traits were not used to determine inflexion points, but to analyze growth coefficients (b-coefficients) during the different phases of larval and juvenile development. The larvae of common dace, ide, chub and common barbel were reared for 30 days at 25 °C in a recirculating system. The initial stocking density of individuals was 85 ind. L-1. Fish were exposed to a 12L:12D photoperiod. Sampled larvae (*n* = 30) were each time anesthetized in a solution of MS-222 (Finquel, Los Angeles, CA, USA) (dose: 0.15 g/dm^3^). The total length (TL) of the fish (±0.01 mm) was measured with AXIO-VISION 4.8.2 software (Zeiss, Jena, Germany) and ProgRes CapturePro 2.5 digital image analysis software (Jenoptic, Germany).

During the experiments, the fish were subjected to morphometric analysis to determine growth rates of selected morphometric traits. These measurements included: head length (HL), trunk length (TRL), tail (post-anal) length (TAL), eye diameter (ED), head depth (HD), body depth (BD), body depth at anus level (BDA) and total length (TL). All measurements were taken along lines parallel or perpendicular to the horizontal axis of the body [14] (Figure 1). Dead, unhatched or abnormal larvae (with malformations) were excluded from the analysis. The allometric growth of each character was expressed as a power function of TL, with the intercept and exponent obtained from linear regressions on log-transformed data. For isometric growth, the growth coefficient was b = 1 for length and b = 3 for weight compared to TL [7,14,15]. Growth analyses were carried out separately in the following three most commonly distinguished phases of the larval period, i.e., preflexion (from hatching to the beginning of flexion of the caudal tip of the notochord); flexion (from flexion of the notochord to the appearance of the hypural plate); postflexion (from the end of flexion to the disappearance of the larval fin fold) and during part of the juvenile period, i.e., from the disappearance of the larval fin fold (beginning of the juvenile period) to the last day of rearing (30 DPH) [3]. These developmental stages were considered achieved when at least 50% of the specimens represented that particular stage. The moment of hatching was determined when 50% of the embryos had left the egg envelopes. In addition, the duration of the different phases and periods, the developmental stage and the total length of the fish at different stages of larval and juvenile life were recorded. The developmental stage of the fish at the time of hatching was determined according to the classification proposed by Peñáz [4].

Differences in mean lengths between the species studied were evaluated using ANOVA and Tukey’s multiple range test for group comparisons. A *p* < 0.05 was considered significant. Statistical analysis was performed using Microsoft Excel and Statistica v. 13.1 (StatSoft Inc., Tulsa, OK, USA).

The research was carried out in accordance with the approval of the local ethics committee for animal experiments (27/2010N for the years 2010–2015).

## 3. Results

### 3.1. Timing of Developmental Phases, Sizes and Developmental Stage of Fish at Different Moments of Larval and Juvenile Life

The duration of the different phases of larval development varied between species. It is worth noting that the differences in the duration of the first two developmental periods, i.e., the beginning and the end of notochord flexion, did not exceed two days. More pronounced differences were observed in the time taken for yolk sac resorption and the disappearance of all larval characteristics (the time of the end of the larval period and the beginning of the juvenile period). For chub and common barbel, body remodeling took 22 days (Table 1). For common dace and ide, it was six days longer and took 28 days. The greatest differences in total length occurred at the beginning of the larval period and decreased with time. At the end of the larval period, fish of all species were of similar size. Their total length was approximately 21 mm and did not differ significantly between species (Table 1). Individuals at the beginning and end of the larval period for each species are shown in Figure 2.

### 3.2. Growth Patterns

The growth coefficients of the different body segments varied during the larval development of the rheophilic species studied. In most of the cases analyzed, the growth rates of the different body segments remained above 1, i.e., positive allometry during the first three developmental phases (exceptions were the lengths “HD” and “BDA” of ide, for which the b values were 0.81 and 0.79, respectively), characterized by significantly lower values in the last period analyzed, where the growth coefficient “b” was close to 1, i.e., near isometry (Table 2). In all the species studied, a slightly different trend was observed in the case of “BD”, where negative growth or negative allometry was observed in the first phase analyzed, and “BDA”, where a clear positive allometry was still observed in the last phase in all the species (the b coefficient took values in the range 1.26–1.44). The middle part of the trunk was also a special case. The TRL showed a clear increase in its growth coefficient during the whole larval period. It started with negative allometry in the preflexion phase (b = 0.32–0.63) and increased to almost isometric growth in the juvenile period (b = 0.95–1) (Table 2).

The differences between species in the growth rates of each morphometric trait were most pronounced in the first post-hatching period (preflexion phase), which coincided with the endogenous feeding phase. As time passed and the ontogenetic development progressed, the values of the b coefficient leveled off. This was particularly noticeable in those parameters where positive allometric growth was recorded for most of the observation period. It is noteworthy that the highest growth rate values in the first growth period analyzed were characteristic of two species, i.e., common barbel and chub. This is particularly evident for parameters such as HL, TAL, HD, ED and BDA (Table 2).

## 4. Discussion

The changes in body shape of the species studied in this paper during the first three life phases analyzed, i.e., preflexion, flexion and postflexion, showed a highly positive allometry in HL and TAL and a negative allometry in TRL. This growth pattern is characteristic of most fish species described to date, not only cyprinids, and is related to the priority given to the growth of body structures associated with food acquisition and movement [11,14,15,16,25]. The rapid growth of the head region (allometric growth of HL, HD, ED) is probably related to the growth and differentiation of the nervous (forebrain, midbrain and hindbrain), sensory (vision and olfaction), respiratory (gill arches and filaments) and digestive systems, since an increase in head size is associated with a more developed nervous system, allowing better oxygen uptake and ingestion of increasingly larger food particles [7,11,26,27,28]. In the posterior part of the body, rapid tail growth (positive allometric growth of TAL, BDA and TD) was accompanied by the development of musculature, unpaired fins, caudal peduncle and fins. Such changes contribute to improved swimming abilities (better detection of zooplankton and avoidance of predators) [7,27,29]. Obviously, there are some differences in this so-called U-shaped growth pattern, which mainly depend on the larvae’s life strategy according to physiological priorities to improve survival rates [30]. These differences may be related to the time of onset and the type of growth. In general, the more developed an organ is, the lower the intensity of its growth. For example, two species from the same temperate climate zone as the species studied in this paper, European perch (*Perca fluviatilis* L.) and burbot (*Lota lota* L.), show rapid growth of the head region, while the caudal part has no initial growth priority and develops isometrically [28,31]. This can be explained by the adaptation to a pelagic life after hatching, where good motor skills are crucial. A similar situation has been observed in the Pacific red snapper *Lutjanus peru* or the seahorse *Hippocampus kuda* Bleeker [32,33].

In the case of the rheophilious cyprinids studied, greater dynamics of change in body proportions can be observed in species that hatch at a lower developmental stage, i.e., chub and especially barbel. In this case, the differences are particularly marked in the growth rates of the head and tail regions. The situation is similar in other cyprinid species such as Caspian shemaya *Alburnus chalcoides* [25], *Schizothorax waltoni* Regan and *Percocypris retrodorslis* [28]. It is also clearly evident in species from other systematic groups. These include species such as *Paralabrax maculatofasciatus* [34], yellowtail kingfish *Seriola lalandi* [30], *Acipenser baeri* [29] and *Paralichthys californicus* [27], where HL and TAL showed a clear positive allometry during the endogenous feeding phase.

It should be noted that the fish species studied were at different stages of development at the time of hatching. During the other periods analyzed, the level of ontogenetic development was identical. In general, cyprinids show a very similar pattern of larval development [4]. Common dace and ide had the same level of development and similar size at hatching. In the case of the other two species, the level of development was identical, although the larvae were less advanced than the two species mentioned above. It should also be noted that the chub was clearly smaller than the common barbel. The chub and common barbel had a significantly larger yolk sac and a less erect body with an obstructed mouth, a lack of pigmentation on the body and a much more developed embryonic fin fold than the ide and dace. Their larvae were less mobile and, unlike common dace and ide, were photophobic, hiding in the shadier areas of the rearing tanks. Ontogenetic development at hatching is species-specific, depends on the reproductive and larval lifestyle and is strongly influenced by environmental conditions, particularly water temperature [35,36,37,38]. Cyprinid species that prefer to incubate at higher water temperatures (e.g., phytophilous carp *Cyprinus carpio* or tench *Tinca tinca*) or that initially exhibit hiding behavior (e.g., litophilous European chub, *Tribolodon hakonensis*, *Schizothorax waltoni* Regan or *Percocypris retrodorslis*) are usually less developed at the time of hatching [15,19,35,39,40,41]. This results in a relatively long resting period between hatching and the first exogenous feeding, during which the fish reach the developmental stage that allows them to become active swimmers. This developmental period is referred to as the free embryo [4,13], yolk sac larvae [3] or compensatory developmental phase [41]. The ontogenic stage of hatched embryos of a given species is characteristically dependent on water temperature. The most advanced fish, with the largest body size and smallest yolk sac, hatch at optimal temperatures. At other temperatures, individuals that leave the egg membranes tend to show less advanced levels of ontogeny [42,43,44]. The increased mobility of embryos in warmer water and the earlier excretion of the hatching enzyme are the reasons why embryos hatch faster at higher temperatures. [36,45,46,47,48].

The reported positive allometric growth of the anterior and posterior parts of the larval body before the development of the trunk region could also be interpreted as an adjustment to the reduction in transport costs, which are rapidly declining with growth. [7,26,49]. Consequently, the rapid growth of the head and tail closely parallels the desirability of reducing drag forces on the body and achieving higher locomotor speeds [10]. Another possible mechanism for reduction of drag in early life stages is the attainment of a fusiform, streamlined body. It is thought that early in life the presence of the pre-anal fin fold provides an energetically inexpensive solution to reducing drag using a minimal amount of tissue [7,50]. With the reduction of the pre-anal fin fold, the body acquires a true fusiform shape through the growth of the gut and the associated coiling of the gut [51]. In the cyprinids studied, growth in the depth of the central body segment (with synchronized elongation of the anterior and posterior segments) is reflected in the positive allometric growth of BD and the strong positive allometric growth of BDA, which was particularly pronounced towards the end of the larval period. In addition, a rapid increase in preorbital length (strong allometric growth of SNL) and an increase in the depth of the occipital region (positive allometric growth of HD) appear to be important in achieving a fusiform, streamlined body shape. A similar growth trend can be observed in other fish species whose survival depends on good swimming skills [26,34,52,53]. For example, less active larvae that exhibit early photophobia and remain close to the bottom for a period of their development (e.g., Siberian sturgeon) or drift (move passively) with water currents (e.g., burbot *Lota lota* L. and Californian halibut) also undergo changes in body proportions that allow drag forces to be reduced by a fusiform body shape, but these are not significant and growth is usually lower [16,28,29,54].

Consistent with the observed priorities, the growth of individual body parts stabilizes during larval development [7,51,55]. In older larvae and juveniles, all growth rates approach 1 (isometric growth) [29,56]. This shift towards isometry, particularly observed in the anterior and posterior body regions, is thought to be a natural transition in growth priorities as basic functions such as feeding and swimming are fulfilled during early development [7]. The moment of allometric growth stabilization occurs at different points in early development; in the above-mentioned species it usually occurs after the postflexion phase and is related to the development of the caudal complex [27,34]. A similar situation occurs in other cyprinids such as *Schizothorax waltoni* Regan and *Percocypris retrodorslis* [53]. In the species studied, the replacement of temporary organs and the appearance of definitive structures at around 21 mm TL coincided with an almost complete reduction in relative growth. Only one of the parameters, BDA, showed clear allometric growth after the fish had reached this length. This is probably a consequence of the attainment of a more streamlined body shape and further ongoing adaptation to life in the river current.

## 5. Conclusions

The results presented in this paper, focusing on the analysis of the dynamics of morphological changes in four riverine cyprinid species during the most dynamic period of fish metamorphosis, confirm the hypothesis put forward in the introduction that growth is prioritized in the most anterior and posterior parts of the body and that its dynamics and duration are influenced by the developmental stage of the hatching larvae and their behavior during successive days of development. These observations are consistent with the growth pattern typical of many fish species, where high allometric growth of head and tail regions is observed at the beginning of the larval period and there is a clear tendency towards isometric growth at the end of the larval period. It was also observed that a lower level of ontogenetic development of the fish after hatching was associated with higher growth rates during the first days after hatching. It should be emphasized that relatively small changes in the growth coefficients of most body segments, manifested by slight differences in the values of the “b” coefficients at successive stages of larval development, were also specific to the species studied. These observations, together with those observed in our previous work, in particular the absence of apparent associated inflexion points in the parameters indicating biphasic growth (changes in growth curves occurring over a wide range of body lengths and ages) and the absence of sharp behavioral and habitat changes (observed under both laboratory and natural conditions), can be considered as gradual adaptation to life in flowing water [14,15,16,17].

## Figures and Tables

**Figure 1 animals-13-02345-f001:**
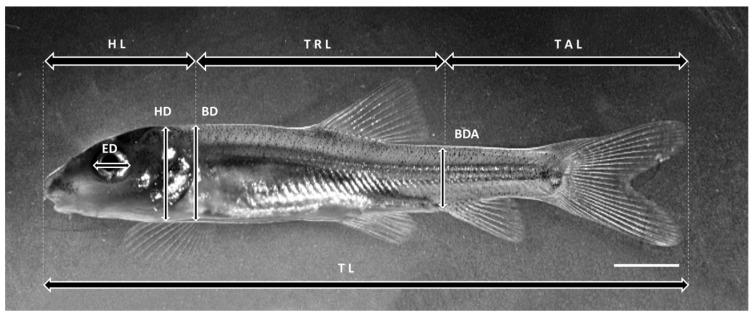
Morphometric characteristics measured in the studied species. ED—eye diameter, HL—head length, TRL—trunk length, TAL—tail length, TL—total length, HD—head depth, BD—body depth, BDA—body depth at anus level. The picture shows common barbel *Barbus barbus* L. Scale bar = 2 mm.

**Figure 2 animals-13-02345-f002:**
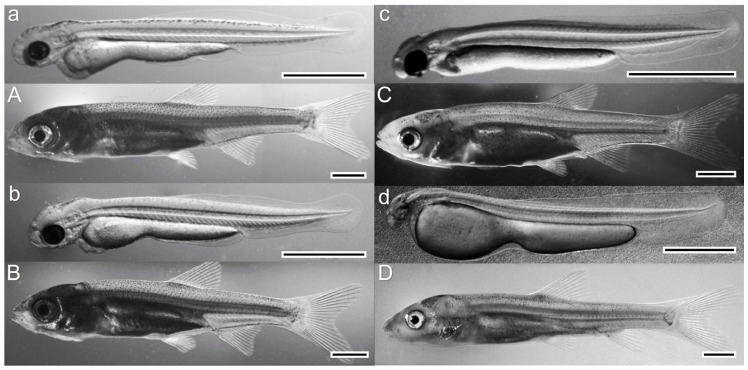
Larval stages of four studied riverine cyprinid species reared at 25 °C: common dace (*Leuciscus leuciscus* L.) (**a**) Yolk-sac larva (0 DPH), (**A**) Juvenile (finfold resorbed, 28 DPH); ide (*Leuciscus idus* L.) (**b**) Yolk-sac larva (0 DPH), (**B**) Juvenile (finfold resorbed, 28 DPH); chub (*Scaulius cephalus* L.) (**c**) Yolk-sac larva (0 DPH) (**C**) Juvenile (finfold resorbed, 22 DPH); common barbel (*Barbus barbus* L.) (**d**) Yolk-sac larva (0 DPH), (**D**) Juvenile (finfold resorbed, 22 DPH). DPH—days post hatching. Scale bars = 2 mm.

**Table 1 animals-13-02345-t001:** Main developmental events of studied species during early development. TL—total length, SD—standard deviation, DPH—days post hatching, DS—developmental stage (according to Penaz [4]). The same letter indexes in the column mean that the values are not statistically different (α = 0.05).

Stages of Early Development												
		Hatching	Yolk Sac Resorption		Start of Notochrd Flexion		End of Notochord Flexion		End of Larval Period (Finfold Resorbed)		Early Juvenile	
	TL ± SD (mm)	(DS)	TL ± SD (mm)	(DPH)	TL ± SD (mm)	(DPH)	TL ± SD (mm)	(DPH)	TL ± SD (mm)	(DPH)	TL ± SD (mm)	(DPH)
Common dace	7.47 ± 0.29 ^a^	ES9 stage ^a^	8.61 ± 0.39 ^a^	3	9.05 ± 0.41 ^a^	4	11.27 ± 1.20 ^a^	10	20.04 ± 1.84 ^a^	28	22.44 ± 1.74 ^a^	30
Ide	7.74 ± 0.18 ^a^	ES9 stage ^a^	8.60 ± 0.23 ^a^	3	8.81 ± 0.18 ^a^	4	11.08 ± 0.48 ^a^	10	20.49 ± 1.24 ^a^	28	21.95 ± 2.02 ^a^	30
Chub	6.68 ± 0.21 ^b^	ES8 stage ^b^	8.25 ± 0.27 ^b^	4	8.45 ± 0.32 ^b^	5	11.97 ± 1.01 ^b^	10	20.84 ± 2.15 ^a^	22	27.95 ± 1.83 ^b^	30
Common barbel	9.53 ± 0.27 ^c^	ES8 stage ^a^	16.10 ± 0.40 ^c^	12	12.44 ± 0.33 ^c^	6	15.41 ± 0.30 ^c^	11	21.38 ± 1.70 ^a^	22	27.16 ± 0.93 ^b^	30

**Table 2 animals-13-02345-t002:** Growth coefficients “b” during the larval and juvenile phases of the 4 cyprinid species. HL, head length; TRL, trunk length; TAL, tail (post-anal) length; ED, eye diameter; HD, head depth; BD; body depth; BDA, body depth at anus level, and total length (TL).

			Developmental Phase		
	Character	Preflexion	Flexion	Postflexion	Juvenile
Common dace (*Leuciscus leuciscus* L.)	HL	1.56	1.37	1.11	0.77
Ide (*Leuciscus idus* L.)		1.62	1.12	1.18	1.00
Chub (*Scaulius cephalus* L.)		1.95	1.25	1.06	0.98
Common barbel (*Barbus barbus* L.)		1.81	1.72	1.14	0.99
Common dace (*Leuciscus leuciscus* L.)	TRL	0.63	0.76	0.61	0.96
Ide (*Leuciscus idus* L.)		0.63	0.74	0.56	1.00
Chub (*Scaulius cephalus* L.)		0.32	0.77	0.68	0.98
Common barbel (*Barbus barbus* L.)		0.38	0.42	0.54	0.95
Common dace (*Leuciscus leuciscus* L.)	TAL	1.15	1.04	1.28	1.05
Ide (*Leuciscus idus* L.)		1.17	1.12	1.31	0.97
Chub (*Scaulius cephalus* L.)		1.47	1.12	1.26	1.02
Common barbel (*Barbus barbus* L.)		1.76	1.45	1.43	1.06
Common dace (*Leuciscus leuciscus* L.)	HD	1.01	1.21	1.34	0.94
Ide (*Leuciscus idus* L.)		0.81	1.4	1.31	1.14
Chub (*Scaulius cephalus* L.)		1.24	1.37	1.21	1.04
Common barbel (*Barbus barbus* L.)		1.73	1.72	1.36	1.04
Common dace (*Leuciscus leuciscus* L.)	BD	−0.56	1.35	1.32	1.11
Ide (*Leuciscus idus* L.)		−1.38	1.35	1.31	1.30
Chub (*Scaulius cephalus* L.)		0.03	1.48	1.26	1.11
Common barbel (*Barbus barbus* L.)		−0.30	1.06	1.29	1.06
Common dace (*Leuciscus leuciscus* L.)	BDA	1.15	1.28	1.78	1.34
Ide (*Leuciscus idus* L.)		0.79	1.18	1.86	1.44
Chub (*Scaulius cephalus* L.)		1.05	1.38	1.76	1.26
Common barbel (*Barbus barbus* L.)		2.85	1.39	1.72	1.31
Common dace (*Leuciscus leuciscus* L.)	ED	1.48	1.18	0.99	0.89
Ide (*Leuciscus idus* L.)		1.23	1.06	1.08	0.99
Chub (*Scaulius cephalus* L.)		1.40	1.07	0.85	0.76
Common barbel (*Barbus barbus* L.)		1.61	0.95	1.04	0.87

## Data Availability

The data presented in this study are available on request from the corresponding author.

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
