# Peer review of "Growth Priorities of the Four Riverine Cyprinids during Early Ontogeny"

_animals, 2023, doi:10.3390/ani13142345_

Round 1

Reviewer 1 Report

Overall the paper is well written and the results are well illustrated. Conclusion needs improvement. 

Author Response

Reviewers' comments:

Reviewer 1

We would like to thank you for your reviews and critical comments on our manuscript. We have carefully considered them all. We agree with most of the comments and believe that their introduction will improve the quality of our manuscript.

General comment/questions: Is there a particular reason for the choice of the 4 fish sp.? Do they represent large fish groups commonly found in EU? Also, in general, for fish that they live in different habitats (fresh water/ sea water ete..), is it expected to have different allometrie growth (due to different growth conditions)?

  • Our response:

The study presented in this paper concerns a group of fishes that can inhabit the same habitat. Their biology has been well studied and described, but despite many similarities in the course of early ontogeny, they are characterised by slightly different biology, e.g. different post-hatching behaviour, as described in the manuscript and our previous work cited. It is also important that the fish described were reared under possibly identical environmental conditions. In addition, it is crucial that these are the only data we have that allow us to accurately determine growth rate values at different life stages.

The genetic factors that determine fish morphology and behaviour are crucial and dominant. They can be masked/altered to some extent by environmental conditions. As the available literature shows, in the case of larvae, this influence is small and does not affect the nature of these changes.

Line 108: As environmental conditions are critical for proper development; I would suggest adding more information in the methods sections (e.g Tm for cach fish type studied etc). Such information will allow the reader to better understand the different growth rates vs developmental stage among species studied

  • Our response:

The methodology presents the main information, noting that detailed information can be found in the cited literature (our previous articles on these species). However, in this chapter, at the suggestion of the reviewer, it has been decided to add some key information on the subject.

Line 109- 110: Add information of Artemia i.e. supplier

  • Our response:

This information was added to the text.

Line 124- 129: Please specify somite number and/ or DPH

  • Our response:

At this age, the fish already have a definitive number of somites. The DPH value for individual fish varied. It is given in the Results section.

Line 140: Add scale bar, specify fish illustrated in picture (for completeness reasons)

  • Our response:

These changes have been made in accordance with the reviewer's suggestion.

Line 151: specify main larvae characteristics

  • Our response:

 In this case it is the duration of the first two periods analysed in the paper, i.e. the beginning and end of the notochord flexion. The description has been clarified.

Lines 164- 170: Use arrows/ arrowheads/etc to better illustrate the organ of discussion

  • Our response:

The reason for including the photos was to show the illustrative morphology of the larvae. In our opinion, the inclusion of additional elements is not advisable as it reduces the legibility of the photos. This information is presented in our detailed larval development paper, which is cited in the manuscript.

Line 257: Add reference.

  • Our response:

The reference to this issue is two lines below (see line 259)

Line 297: In the conclusion, the importance of findings should be stressed out. Overall, this section should be improved, State the most important outcome(s) of the research. Please avoid summarising points already mentioned

It's important to interpret your outcomes/ findings at a higher level of abstraction (show the *big picture'- importance of your work). Please add a phrase showing that you accomplished the aim stated in your introduction.

  • Our response:

This section of the text has been amended as suggested by the reviewer.

„The results presented in this paper, focusing on the analysis of the dynamics of morphological changes in four reophilic cyprinid species during the most dynamic period of fish metamorphosis, confirm the hypothesis put forward in the introduction that growth is prioritized in the most anterior and posterior parts of the body and that its dynamics and duration are influenced by the developmental stage of the hatching larvae and their behavior during successive days of development. These observations are consistent with the growth pattern typical of many fish species, where high allometric growth of the head and tail is observed at the beginning of the larval period and a clear tendency towards isometric growth at the end of the larval period. It should be emphasised that relatively small changes in the growth coefficients of most body segments, manifested by slight differences in the values of the "b" coefficients at successive stages of larval development, were also specific to the species studied. These observations, together with those observed in our previous work, in particular: the absence of apparent associated inflection points in the parameters indicating biphasic growth (changes in growth curves occurring over a wide range of body lengths and ages) and the absence of sharp behavioral and habitat changes (observed under both laboratory and natural conditions), can be considered as a gradual adaptation to life in flowing water [14-17].”

Reviewer 2 Report

Review of Kupren et al (2023), “Growth priorities of the four riverine cyprinids during early ontogeny,” for consideration in Animals (MDPI).

OVERALL IMPRESSION

In this study, Kupren et al describe the growth patterns of four European fish species during larval and early juvenile development.  Their focus is on external (superficial) growth of clearly identifiable structures and body regions, exclusively during post-hatching stages.  Therefore, their purpose isn’t an inter-species comparison of equivalent developmental stages, but rather to make sense of the differences in growth patterns in light of species-specific behaviors and environmental conditions.  The authors perform a straightforward, solid analysis of these parameters and make reasonable interpretations of the data which highlight the fine-tuned correlation between development and behavior.  I have only minor concerns and/or requests for clarification and would recommend this paper for publication if they can be satisfactorily resolved.

MINOR CONCERNS (in order of occurrence)

1.     Introduction: My understanding is that genera are always italicized, but here they are not italicized when mentioned alone (lines 63-66).  Please verify that this is appropriate convention.  Also, Artemia nauplii should be italicized (lines 109-110).

2.     Methods:  Since larvae were reared in optimal lab conditions, I would expect the time of hatching within each group to be somewhat synchronous.  However, I know from studies in zebrafish that this isn’t always the case (i.e., most hatch within 12 hours of each other but several may not hatch until the following day even with normal morphology).  Were unhatched larvae manually dechorionated and included in the samples, or were they excluded?  It would be helpful to clarify.  You already mentioned that dead or deformed embryos were excluded, but delayed hatching isn’t strictly a result of abnormal morphology.

3.     Methods:  I assume the larvae were briefly anesthetized in order to photograph.  If so, how?

4.     Methods:  I cannot find any mention of sample size in the paper.  This would be helpful to include as part of the statistical analysis.

5.     Methods, line 124: “tree” should be “three”.

6.     Results, Table 2: “Juvenile” is misspelled in the header row.  The second mention of “Common barbel” is not capitalized.  Some data only include one digit after the decimal instead of two.

7.     Discussion, line 206: Why is the forebrain not mentioned here (only the midbrain/hindbrain) given that the forebrain is by far the largest component of the developing nervous system?

8.     Discussion, lines 221-222: References 29 and 32 don’t have brackets around them.  A period should be included after the word “crucial.”

Author Response

Reviewers' comments:

Reviewer 2

Many thanks to Reviewer 2 for all the constructive comments and recommendations.

 The specific comments made by Reviewer 2 are addressed below:

Review of Kupren et al (2023), “Growth priorities of the four riverine cyprinids during early ontogeny,” for consideration in Animals (MDPI).

OVERALL IMPRESSION

In this study, Kupren et al describe the growth patterns of four European fish species during larval and early juvenile development.  Their focus is on external (superficial) growth of clearly identifiable structures and body regions, exclusively during post-hatching stages.  Therefore, their purpose isn’t an inter-species comparison of equivalent developmental stages, but rather to make sense of the differences in growth patterns in light of species-specific behaviors and environmental conditions.  The authors perform a straightforward, solid analysis of these parameters and make reasonable interpretations of the data which highlight the fine-tuned correlation between development and behavior.  I have only minor concerns and/or requests for clarification and would recommend this paper for publication if they can be satisfactorily resolved.

MINOR CONCERNS (in order of occurrence)

  1. Introduction: My understanding is that genera are always italicized, but here they are not italicized when mentioned alone (lines 63-66).  Please verify that this is appropriate convention.  Also, Artemia nauplii should be italicized (lines 109-110).

  • Our response:

These comments have been taken into account in the text.

  1. Methods: Since larvae were reared in optimal lab conditions, I would expect the time of hatching within each group to be somewhat synchronous.  However, I know from studies in zebrafish that this isn’t always the case (i.e., most hatch within 12 hours of each other but several may not hatch until the following day even with normal morphology).  Were unhatched larvae manually dechorionated and included in the samples, or were they excluded?  It would be helpful to clarify.  You already mentioned that dead or deformed embryos were excluded, but delayed hatching isn’t strictly a result of abnormal morphology.

  • Our response:

In fact, it takes a certain amount of time for the larvae to hatch, and this depends most on the incubation temperature. Measurements of the larvae at hatching and at other times were analysed when half of the individuals had reached a given developmental stage or had left the egg envelopes. This was the procedure used in our previous articles analysing early development. This information was presented in the Materials and methods chapter.

Dead, unhatched or abnormal larvae (with malformations) were excluded from the analysis.

Relevant information has been included in the text.

  1. Methods: I assume the larvae were briefly anesthetized in order to photograph.  If so, how?
  2. Methods: I cannot find any mention of sample size in the paper.  This would be helpful to include as part of the statistical analysis.

  • Our response:

Sampled larvae (n = 30) were each time anesthetized in a solution of MS-222 (Finquel, Los Angeles, CA, USA) (dose: 0.15 g/dm3 ). The total length (TL) of the fish (±0.01 mm) was measured with AXIO-VISION 4.8.2 software (Zeiss, Jena, Germany) and ProgRes CapturePro 2.5 digital image analysis software (Jenoptic, Germany).

Relevant information has been included in the text.

  1. Methods, line 124: “tree” should be “three”.

  • Our response:

Suggested changes have been made to the text.

  1. Results, Table 2: “Juvenile” is misspelled in the header row. The second mention of “Common barbel” is not capitalized.  Some data only include one digit after the decimal instead of two.

  • Our response:

Suggested changes have been made to the text.

  1. Discussion, line 206: Why is the forebrain not mentioned here (only the midbrain/hindbrain) given that the forebrain is by far the largest component of the developing nervous system?

  • Our response:

We agree with the reviewer that the information is incomplete. Suggested changes have been made to the text.

  1. Discussion, lines 221-222: References 29 and 32 don’t have brackets around them. A period should be included after the word “crucial.”

  • Our response:

Suggested changes have been made to the text.
